# Effects of di-(2-ethylhexyl) phthalate on Transcriptional Expression of Cellular Protection-Related *HSP60* and *HSP67B2* Genes in the Mud Crab *Macrophthalmus japonicus*

**Kiyun Park [1], Won-Seok Kim [2] and Ihn-Sil Kwak [1,2,\*]**

1    Fisheries Science Institute, Chonnam National University, Yeosu 59626, Korea; ecoblue@hotmail.com
2    Faculty of Marine Technology, Chonnam National University, Yeosu 59626, Korea; csktjr123@gmail.com
\*    Correspondence: iskwak@chonnam.ac.kr; Tel.: +82-61-6597148; Fax: +82-61-6597149

**Abstract:** Di-2-ethylhexyl phthalate (DEHP) has attracted attention as an emerging dominant phthalate contaminant in marine sediments. *Macrophthalmus japonicus*, an intertidal mud crab, is capable of tolerating variations in water temperature and sudden exposure to toxic substances. To evaluate the potential effects of DEHP toxicity on cellular protection, we characterized the partial open reading frames of the stress-related heat shock protein 60 (*HSP60*) and small heat shock protein 67B2 (*HSP67B2*) genes of *M. japonicus* and further investigated the molecular effects on their expression levels after exposure to DEHP. Putative *HSP60* and small *HSP67B2* proteins had conserved HSP-family protein sequences with different C-terminus motifs. Phylogenetic analysis indicated that *M. japonicus HSP60* (*Mj-HSP60*) and *M. Japonicus HSP67B2* (*Mj-HSP67B2)* clustered closely with *Eriocheir sinensis HSP60* and *Penaeus vannamei HSP67B2*, respectively. The tissue distribution of Heat shock proteins (HSPs) was the highest in the gonad for *Mj-HSP60* and in the hepatopancreas for *Mj-HSP67B2*. The expression of *Mj-HSP60* Messenger Ribonucleic Acid (mRNA) increased significantly at day 1 after exposure to all doses of DEHP, and then decreased in a dose-dependent and exposure time-dependent manner in the gills and hepatopancreas. *Mj-HSP67B2* transcripts were significantly upregulated in both tissues at all doses of DEHP and at all exposure times. These results suggest that cellular immune protection could be disrupted by DEHP toxicity through transcriptional changes to HSPs in crustaceans. Small and large HSPs might be differentially involved in responses against environmental stressors and in detoxification in *M. japonicus* crabs.

**Keywords:** di(2-ethylhexyl) phthalate (DEHP); crustacean; heat shock proteins (HSPs); gene expression; environmental risk assessment

## 1. Introduction

Artificial chemical additives have come to the fore as one of the main environmental pollution triggers. Plasticizers, which assign flexibility and durability to plastic, have been heavily utilized, owing to the widespread application of plastic products. As the most common plasticizer, di-2-ethylhexyl phthalate (DEHP) has contributed to the manufacture of flexible products from solid plastics such as polyvinyl chloride [1]. Owing to its widespread use, DEHP is ubiquitously released into the aquatic environment [2,3]. A recent study showed that the main source of DEHP is emissions from household sewage and sludge disposal activities [2]. DEHP is detected at high levels in all sediment samples taken from coastal bays, indicating ubiquitous contamination of the marine environment [3]. DEHP concentrations were found to range from 3020 to 3970 ng/g in sediments from the Kuwait Coast, Pearl River Delta in China, and Kaohsiung Harbor in Taiwan [4–6]. In addition, in the northwestern

Mediterranean Sea, the range of DEHP concentrations was 42–802 ng L$^{-1}$ and 130–924 ng L$^{-1}$ in the surface seawater (depth 0.5 m) and bottom seawater (depth 30 m), respectively [7]. DEHP concentrations were found to range from 62 to 4352 ng L$^{-1}$ from the bottom to the surface seawater of the Bohai Sea and the Yellow Sea, China [8]. DEHP, an endocrine-disrupting chemical (EDC), exhibits a perturbing effect on steroidogenesis activities [9,10].

*Macrophthalmus japonicus* is one of the main benthic species ubiquitously detected in tidal flats and shows high distribution rates in estuarine regions of Korea and Japan [11,12]. As a main member of the tidal flat food chain, this species contributes to the maintenance of biodiversity in estuarine ecosystems. Because of their dominant distribution, crabs might be a good candidate organism to sense changes in the condition of the surrounding environment, as well as changes involving food reserves, as they have abundant nutrients and are of high economic value in commercial fisheries. However, crab habitats are easily exposed to great hazards, such as plastic waste pollutants and chemicals that are transported into mud flats through rivers or from the ocean. The effects of various stress conditions, such as salinity and heavy metal and biocide contaminants, have been reported following expression analysis of immune-related or stress-related genes in crabs [11,13–16]. A recent study showed the relationships between EDCs and gene expression alterations involving crab innate immune systems [17], but there have been no studies of the relationship between stress-related gene expression and EDC exposure. Despite its biological importance as a nutritional resource, few studies have been conducted on the *M. japonicus* genomic DNA sequence.

Heat shock proteins (HSPs) are ubiquitous proteins secreted in cells after exposure to stressful conditions and are classified into six major groups (*HSP27, HSP60, HSP70, HSP90*, and large HSPs) based on their molecular weights [18,19]. HSPs function as molecular chaperones to prevent the formation of denatured proteins during high temperature stress and exhibit upregulation in their expression patterns under such stress conditions [18,20]. In addition, these stress proteins play an important role in the maintenance of normal polypeptide structures and in the promotion of correct refolding of cellular proteins in response to various external stimuli, such as anoxia, heavy metals, or chemicals, which cause protein denaturation [20–22]. HSPs assist in protecting cellular homeostasis from such stress. *HSP60* is well known as a pro-apoptotic molecule, which induces apoptosis and acts as a chaperone for proteins transcribed from mitochondrial DNA [23–25]. *HSP60* is a highly immunogenic protein, which is implicated in a variety of autoimmune diseases [26,27]. The upregulation of *HSP60* indicates its involvement in crucial functions mediating immune responses in the Chinese mitten crab, *Eriocheir sinensis*, after crustacean pathogen infection [27]. *HSP67B2* was characterized as a Relish-regulated gene in the innate immunity of the Chinese shrimp (*Fenneropenaeus chinensis*) [28]. However, there is limited information about the molecular characterization and expression responses involving the crustacean *HSP67B2*.

In the present study, we identify two stress-related genes, *Mj-HSP60* and *Mj-HSP67B2*, in *M. japonicus* crabs to evaluate the toxic effects of DEHP on cellular immune protection in crustaceans. We investigate the genomic structure, phylogenetic relationships with other homologous HSPs, and transcriptional responses of HSPs under DEHP stress. We seek to provide molecular information regarding the influence of EDCs on stress-related gene expression in *M. japonicus*.

## 2. Materials and Methods

### 2.1. Ethical Statement

All experiments involving *M. japonicus* crabs in this study were carried out in accordance with the guidelines and regulations approved by the Institutional Animal Care and Use Committee of Chonnam National University.

### 2.2. Preparation of M. japonicus Individuals

Crabs used in this study were collected from the Yeosu marine products market in Korea. All individuals involved were $3 \pm 0.5$ cm in shell height, $3.5 \pm 0.8$ cm in shell width, and $7.5 \pm 3.5$ g in body weight. We prepared glass tanks ($45.7 \times 35.6 \times 30.5$ cm) filled with seawater at 18 °C, with 25% salinity and a photoperiod of 12 h. Crabs were stabilized in glass tanks for 1 day prior to exposure to DEHP solutions. After 1 day, healthy, undamaged crabs were selected for DEHP exposure experiments (below).

### 2.3. DEHP Exposure Experiments

DEHP solutions were made from a solid compound (99%, Junsei Chemical Co. Ltd., Tokyo, Japan). For preparation of a 10 mg $L^{-1}$ stock solution of DEHP, we dissolved DEHP in 99% acetone at room temperature. This stock solution was diluted with seawater for DEHP solutions with concentrations of 1, 10, and 30 $\mu$g $L^{-1}$. A concentration of <0.5% acetone was used as a solvent control. For the DEHP exposure experiments, a total of 40 crabs were randomly divided into four experimental groups (1, 10, and 30 $\mu$g $L^{-1}$ DEHP solutions and solvent control). Ten crabs were placed in each glass tank and exposed to one of the three doses of DEHP over days 1, 4, and 7, respectively. Three individuals were selected for tissue extraction at each time interval from the DEHP treatment and control groups. Food was not provided for the crabs, but seawater with equivalent concentrations of DEHP was added every day during the experiments. The experiments were conducted in triplicate with independent samples.

### 2.4. Total RNA Extraction and cDNA Synthesis

Crab gill and hepatopancreatic tissues were acquired from the exposure and control groups. Total RNA was extracted using TRIzol reagent (Life Technologies, Rockville, MD, USA) with Recombinant DNase I (Takara, Otsu, Japan) according to the manufacturers' protocols. The concentration of each RNA sample was measured using a Nano-Drop 1000 (Thermo Fisher Scientific, Waltham, MA, USA). RNA integrity was checked by 1% agarose gel electrophoresis. Single-stranded Complementary Deoxyribonucleic Acid (cDNA) synthesis was carried out with 1000 ng of total RNA using an oligo dT primer (50 $\mu$M) for reverse transcription in 20 $\mu$L reactions (PrimeScript™ 1st strand cDNA synthesis kit, Takara) according to the manufacturer's protocol.

### 2.5. Gene Expression Analysis Using Quantitative Reverse-Transcription PCR (RT-PCR) Amplification

To confirm the expression patterns of *Mj-HSP60* and *Mj-HSP67B2* in various tissues of *M. japonicus*, and in the control and DEHP-exposed samples, quantitative RT-PCR was carried out on an ExicyclerTM96 instrument (Bioneer, Daejeon, Korea). Each reaction was conducted in a final volume of 20 $\mu$L containing 10 $\mu$L of Accuprep®2 × Greenstar qPCR Master Mix (Bioneer, Daejeon, Korea), 6 $\mu$L of DEPC-treated water, 0.5 $\mu$L each of sense primer and antisense primer (10 pM), and 3 $\mu$L of 30-fold diluted cDNA sample as a template. Quantitative RT-PCR of two genes was carried out for 40 cycles of 95 °C for 15 s and 60 °C for 45 s using the following primer pairs: *Mj-HSP60* forward 5′-CCCTGAAGGATGAGCTTGAG-3′; *Mj-HSP60* reverse 5′-GCTGGGATGATGGA CTGAAT-3′; *Mj-HSP67B2* forward 5′-GAGCCGCGGTAGATTCTAT G-3′; *Mj-HSP67B2* reverse 5′-CTGGACAAGGAGGGTTTCAA-3′; Glyceraldehyde-3-Phosphate Dehydrogenase (GAPDH) forward 5′-TGCTGATGCACCCATGTTT G-3′; and *GAPDH* reverse 5′-AGGCCCTGGACAATCTCAA AG-3′. Melting curves were determined by increasing the temperature from 68 °C to 94 °C. All samples were amplified in triplicate to ensure reproducibility. The relative expression level of each transcript was determined using *M. japonicus GAPDH* as an internal reference gene and employing the $2^{-\Delta\Delta Ct}$ method [29].

## 2.6. M. japonicus Hsp Identification and Bioinformatics Analysis

Two HSP genes (*Mj-HSP60* and *Mj-HSP67B2*) were identified by screening a previously generated 454 GS-FLX transcriptome database. Sequences were analyzed based on nucleotide and protein databases using the BLASTN and BLASTX programs (National Center for Biotechnology Information, U.S. National Library of Medicine, Bethesda, MD, USA), respectively [30]. Two domains, the chaperonin-like super family of *Mj-HSP60* and Rhodonase (RHOD) superfamily of *Mj-HSP67B2*, were identified by PROSITE profile analysis [31]. A phylogenetic tree for the two HSPs was generated by the neighbor joining method using Molecular Evolutionary Genetic Analysis (MEGA X, Pennsylvania State University, State College, PA, USA) [32] with 1000 bootstrap replications.

## 2.7. Statistical Analysis

The Statistical Package for the Social Sciences (SPSS) 12.0 KO (SPSS Inc., Chicago, IL, USA) was used for statistical analysis in this study. Data are presented as the mean ± standard deviation. Two-way analysis of variance was conducted to identify the statistical effects of the exposure period and each DEHP dose on *Mj-HSP60* and *Mj-HSP67B2* mRNA expression. Significant differences were presented as $*P < 0.05$ and $**P < 0.01$.

## 3. Results

### 3.1. Characterization of Mj-HSP60 and Mj-HSP67B2 in M. japonicus

We identified two HSP genes (*Mj-HSP60* and *Mj-HSP67B2*) in our 454 GS-FLX transcriptome analysis [33] that were composed of 1360 nucleotides (nt) and 511 nt, which comprised open reading frames encoding 330 and 149 amino acids, respectively (Figures 1A and 2A). *Mj-HSP60* encoded a mature protein of 330 amino acids, 75 bp of 5′ untranslated region (UTR) and 57 bp of 3′ UTR, with a putative methionine initiation codon (ATG) beginning at 58 nt and a stop codon ending at 1224 nt. The SignalP Server (ExPASy) [34] predicted that the first 28 amino acids in the N-terminal region of the polypeptide chain would form a signal peptide sequence. We found that *Mj-HSP60* included a chaperonin-like super family main domain, whereas a RHOD superfamily motif was detected in *Mj-HSP67B2* (Figure 2A). The predicted molecular mass of the deduced amino acid sequence was 61 kDa, with an estimated isoelectric point (pI) of 5.74. *Mj-HSP60* was identified by a BLAST search of the National Center for Biotechnology Information (NCBI) non-redundant (nr) database. To understand the evolutionary position of the *Mj-HSP60*, we undertook phylogenetic analysis using another 11 species of crustaceans. As shown in Figure 1B, the phylogenetic tree consisted of two clades involving 12 crustacean species. The *Mj-HSP60* formed one main clade with other crabs (*Eriocheir sinensis, Scylla paramamosain,* and *Portunus trituberculatus*) and crayfish (*Cherax cainii*, *Cherax quadricarinatus*, and *Cherax destructor*). The other clade was composed of shrimp species (*Macrobrachium nipponense, Macrobrachium rosenbergii, Penaeus japonicus, Penaeus monodon,* and *Penaeus vannamei*). For clear annotation of *Mj-HSP67B2*, we examined the RHOD superfamily domain sequence (98 amino acids) using BLASTN searches of the nr database to detect sequences of other species with high similarity. We carried out pairwise alignment of *Mj-HSP67B2* using EMBOSS alignment (EMBL-EBI, Cambridgeshire, UK) [35] with sequences identified in BLAST searches. The results showed 35.9–72.8% sequence identity, 54.4–81.6% similarity, and 4.9–10.5% gap percentage when compared with *HSP67B2* from other species (Table 1). The *Mj-HSP67B2* sequence revealed considerable identity (72.8%), similarity (81.6%), and gap percentage (4.9%) with *Penaeus vannamei HSP67B2*. In addition, phylogenetic analysis of the *Mj-HSP67B2* was carried out using data from various arthropod species, owing to deficient genomic information regarding the *HSP67B2* in crustaceans (Figure 2B). The results showed that the two main clades were divided into Crustacea and Insecta, including mosquito and fly species. The *Mj-HSP67B2* showed the closest phylogenetic relationship to *Penaeus vannamei HSP67B2*. Given these results from analysis of phylogenetic and pairwise sequence alignment comparisons, our transcript sequence from the transcriptome database was identified as *Mj-HSP67B2*.

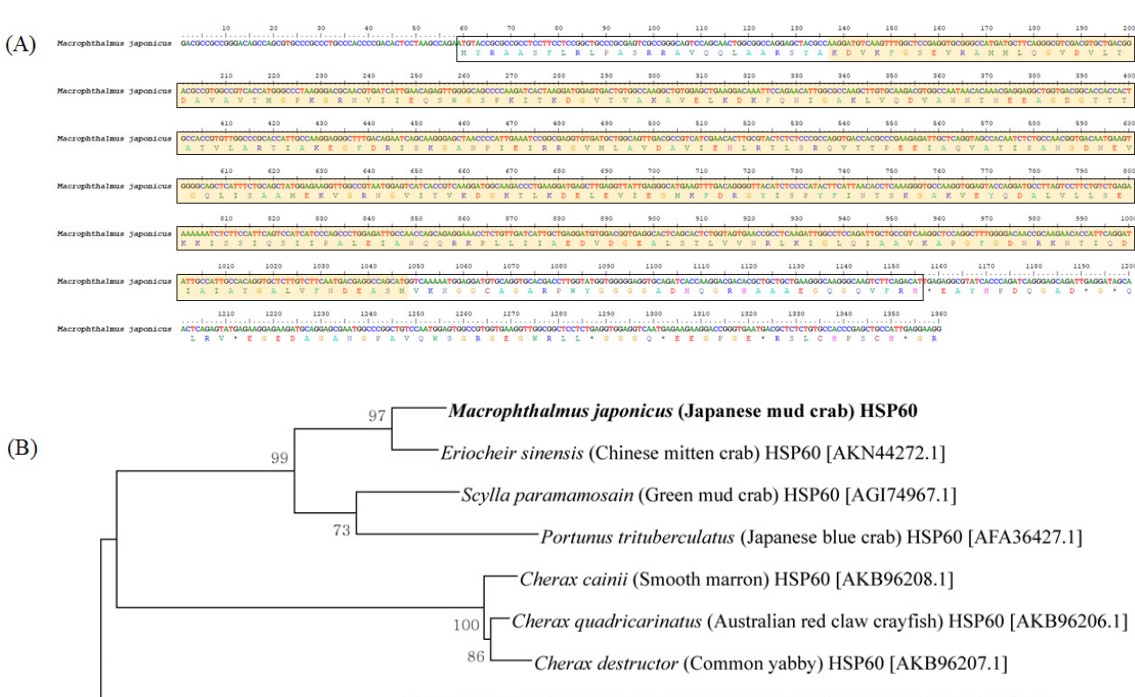

**Figure 1.** Genomic information of *Macrophthalmus japonicus HSP60* sequences identified in this study. (**A**) *Mj-HSP60* structure was represented using the BioEdit program (North Carolina State University, Raleigh, NC, USA). The open reading frame (ORF) of *Mj-HSP60* was predicted using the ExPASy tool and is shown as a black box. The yellow box indicates the chaperonin-like super family domain. (**B**) Phylogenetic analysis of *Mj-HSP60* with known *HSP60* sequences from 11 Crustacean species. The phylogenetic tree is based on amino acid sequences translated from *Mj-HSP60* ORF by the neighbor joining method (bootstrap value 1000) using MEGA X software. The numbers at the nodes represent the bootstrap majority consensus values for 1000 replicates. GenBank accession numbers are shown with scientific and common names of each species.

### 3.2. Expression Analysis of Mj-HSP60 and Mj-HSP67B2 in Various Tissues of M. japonicus

To better understand the expression patterns of *Mj-HSP60* and *Mj-HSP67B2*, quantitative RT-PCR was carried out for six tissue sources (gill, hepatopancreas, muscle, gonad, heart, and stomach) of *M. japonicus*. The highest level of *Mj-HSP60* expression was found in the gonad, while *Mj-HSP67B2* was predominantly expressed in the hepatopancreas (Figure 3). In the gonad, *Mj-HSP60* was expressed 3.7-fold higher than *Mj-HSP67B2*. In contrast, *Mj-HSP67B2* exhibited a higher expression level than *Mj-HSP60* in the gills (1.7-fold) and hepatopancreas (3.1-fold). Relatively low levels of *Mj-HSP60* and *Mj-HSP67B2* expression were observed in the muscle, heart, and stomach tissues.

(A)

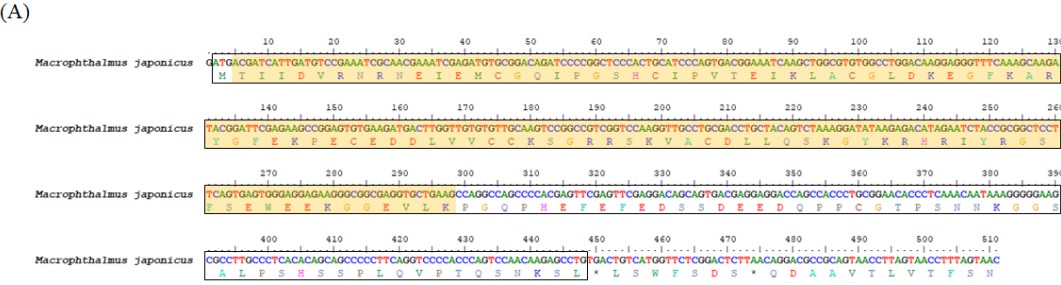

(B)

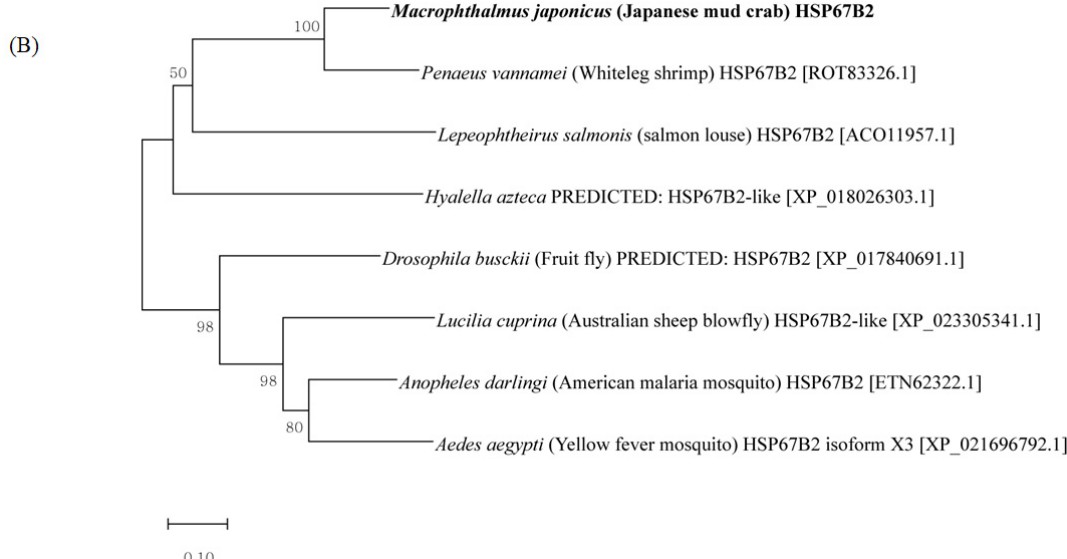

**Figure 2.** Sequence information for *Macrophthalmus japonicus HSP67B2* identified in this study. (**A**) An open reading frame (ORF) of *Mj-HSP67B2* was predicted using the ExPASy tool and is represented by a black box. The yellow box indicates a RHOD superfamily domain. (**B**) Phylogenetic analysis of *Mj-HSP67B2* with known *HSP67B2* sequences from seven Arthropoda species. The phylogenetic tree is based on amino acid sequences translated from *Mj-HSP67B2* ORF by the neighbor joining method (bootstrap value 1000) using MEGA X software. The numbers at the nodes represents the bootstrap majority consensus values for 1000 replicates. GenBank accession numbers are shown with the scientific and common names of each species.

**Table 1.** Percentage identity, similarity, and gaps involving *Macrophthalmus japonicus* HSP67B2 and HSP67B2 homologs from other species at the amino acid level

| Species | Gene Name | Accession Number | RHOD Superfamily Domain Length | Identity (%) | Similarity (%) | Gap (%) |
|---|---|---|---|---|---|---|
| *Macrophthalmus japonicus* | Heat Shock protein 67B2 | | 98 | | | |
| *Penaeus vannamei* | Heat Shock protein 67B2 | ROT83326.1 | 103 | 72.8 | 81.6 | 4.9 |
| *Lepeophtheirus salmonis* | Heat Shock protein 67B2 | ACO11957.1 | 106 | 43.4 | 60.4 | 7.5 |
| *Lucilia cuprina* | Heat Shock protein 67B2-like | XP_023305341.1 | 101 | 40.0 | 60.0 | 10.5 |
| *Hyalella azteca* | PREDICTED: heat shock protein 67B2-like | XP_018026303.1 | 106 | 44.3 | 58.5 | 7.5 |
| *Drosophila busckii* | PREDICTED: heat shock protein 67B2 | XP_017840691.1 | 99 | 39.8 | 58.3 | 8.7 |
| *Anopheles darlingi* | Heat Shock protein 67B2 | ETN62322.1 | 103 | 39.8 | 56.3 | 4.9 |
| *Aedes aegypti* | Heat Shock protein 67B2 isoform X3 | XP_021696792.1 | 99 | 35.9 | 54.4 | 8.7 |

Pairwise identity percentage was calculated using the EMBOSS alignment program.

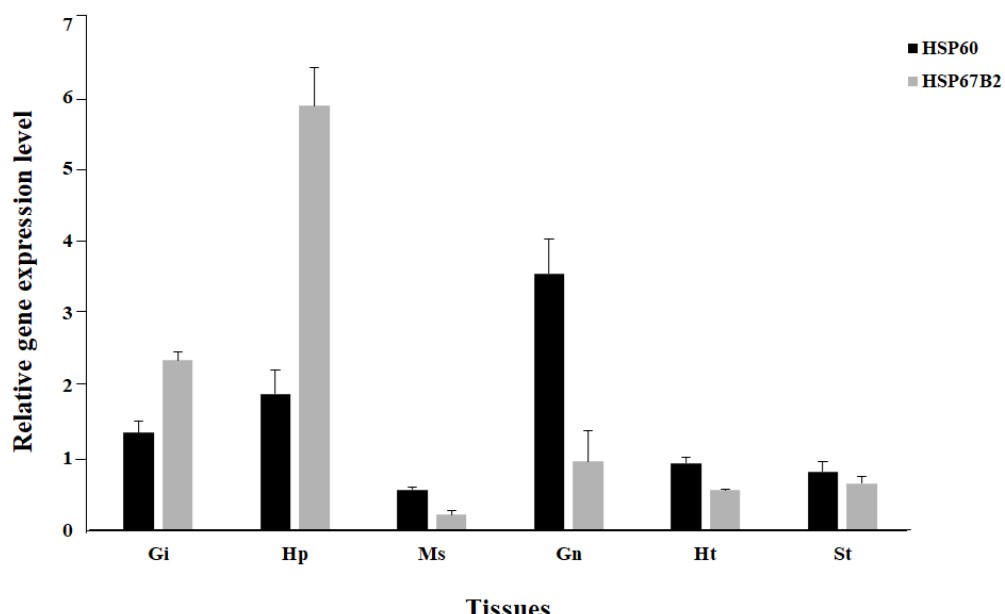

**Figure 3.** Relative mRNA expression levels of *HSP60* and *HSP67B2* in various *Macrophthalmus japonicus* tissues. Six tissues were used in this experiment. Quantitative reverse-transcription (RT)-PCR was conducted in triplicate. Bars indicate the standard deviation of the mean. mRNA expression was normalized against *GAPDH*. Abbreviations: Gill (Gi), Hepatopancreas (Hp), Muscle (Ms), Gonad (Gn), Heart (Ht), and Stomach (St).

### 3.3. M. japonicus Mj-HSP60 Expression Changes after DEHP Exposure

To confirm the effects of DEHP exposure on *Mj-HSP60* expression, we conducted quantitative RT-PCR analysis using mRNA acquired from the gill and hepatopancreas samples after exposure to DEHP for 1, 4, and 7 days. *Mj-HSP60* was expressed approximately 8.2-fold higher after exposure to 1 µg $L^{-1}$ DEHP ($P < 0.01$), 3.2-fold higher for 10 µg $L^{-1}$ ($P < 0.05$), and 9.4-fold higher for 30 µg $L^{-1}$ ($P < 0.01$) in the gill tissue on day 1 (Figure 4A). With the passage of time, expression levels gradually decreased in all DEHP concentration groups. By day 4, for the 10 and 30 µg $L^{-1}$ treatment groups, expression levels were restored to control levels. By day 7, *Mj-HSP60* expression levels were lower than those of the control. In particular, sharp decreases in expression levels were found in 10 µg $L^{-1}$ (0.3-fold) and 30 µg $L^{-1}$ (0.21-fold) ($P < 0.05$) groups. In the hepatopancreatic tissue, expression levels of *Mj-HSP60* exhibited an overall increased pattern compared to the expression levels in the controls on day 1 (Figure 4B). Expression levels significantly increased by 2.4-fold for 1 µg $L^{-1}$, 2.6-fold for 10 µg $L^{-1}$, and 2.9-fold for 30 µg $L^{-1}$ DEHP ($P < 0.05$). By days 4 and 7, *Mj-HSP60* expression levels returned to control levels for the 1 µg $L^{-1}$ group. In the 10 µg $L^{-1}$ DEHP group, *Mj-HSP60* expression decreased to <0.5-fold on day 4, and then recovered slightly toward that of control levels by day 7.

### 3.4. Variation in Expression of Mj-HSP67B2 after DEHP Exposure in M. japonicus

Expression of *Mj-HSP67B2* consistently increased in the gill and hepatopancreatic tissues for 4 days after DEHP exposure at all concentrations (Figure 5). After a peak in expression at day 4, *Mj-HSP67B2* levels somewhat decreased. These *Mj-HSP67B2* expression patterns were found in the two tissues, regardless of DEHP exposure concentration. Although expression levels of *Mj-HSP67B2* decreased after day 4, the expression was still maintained in the gill tissue at higher levels than those of the controls for all concentration groups, except on day 7 (0.86-fold) for the 1 µg $L^{-1}$ group (Figure 5A). Similar changes in *Mj-HSP67B2* expression levels were noted in the hepatopancreas tissue. *Mj-HSP67B2* was strongly overexpressed for 4 days in response to exposure to all concentrations of DEHP ($P < 0.05$), and its expression levels displayed dose-dependent and time-dependent increases for

4 days (Fig. 5B). The highest expression levels were noted on day 4 in each DEHP concentration group (3.9-fold for 1 μg L$^{-1}$ ($P < 0.05$), 5.48-fold for 10 μg L$^{-1}$ ($P < 0.01$), and 5.88-fold for 30 μg L$^{-1}$ ($P < 0.01$).

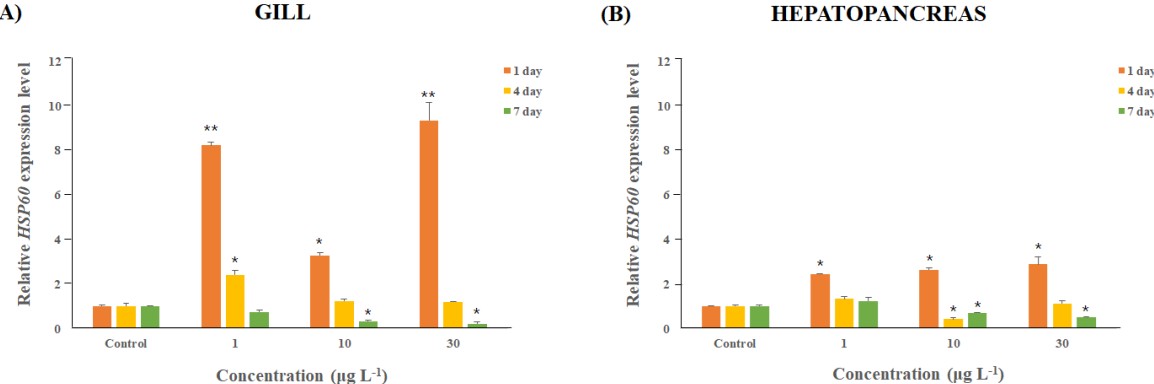

**Figure 4.** Expression analysis of *HSP60* in the (**A**) gill and (**B**) hepatopancreas of *Macrophthalmus japonicus* exposed to 1, 10, and 30 μg L$^{-1}$ DEHP after 1, 4, and 7 days. Values were normalized against *GAPDH*. Bars indicate the standard deviation of the mean. Statistically significant differences are represented by asterisks as $^*P < 0.05$ and $^{**}P < 0.01$, compared to controls (control ratio value = 1).

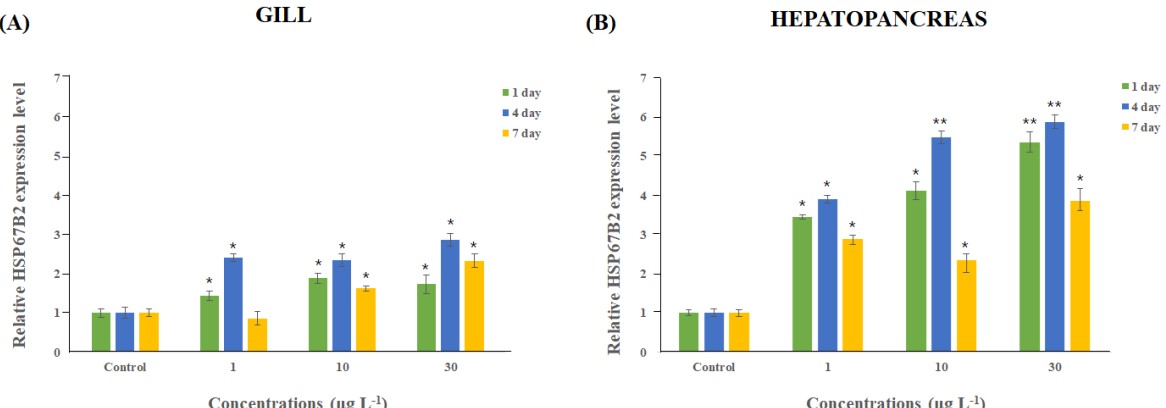

**Figure 5.** Expression analysis of *HSP67B2* in the (**A**) gill and (**B**) hepatopancreas of *Macrophthalmus japonicus* exposed to 1, 10, and 30 μg L$^{-1}$ DEHP for 1, 4, and 7 days. The values were normalized against *GAPDH*. Bars indicate the standard deviation of the mean. Statistically significant differences are represented by asterisks as $^*P < 0.05$ and $^{**}P < 0.01$ as compared to controls (control ratio value = 1).

## 4. Discussion

Cellular responses to stressors are an evolutionary, ubiquitous, and essential mechanism for cell survival. HSPs are known as extrinsic chaperons that are involved in certain cellular processes, such as germ cell differentiation, reproduction, development, thermoprotection, mammalian autoimmune defense, and toxic stress responses, and they have even been regarded as a potential marker of environmental stress [36–42]. HSPs are found in all eukaryotes and are identified based on their size, molecular weight, and functions. *HSP60*, *HSP70* and *HSP90* are highly conserved genes and are stress-inducible and multigenic [43]. It has been observed that the *HSP60* and *HSP70* family members play significant roles in cell survival, stress, and thermal tolerance in response to various heat shocks [44].

Here, we studied two stress-related genes, *Mj-HSP60* and *Mj-HSP67B2*, and conducted expression analysis in different tissues of *M. japonicus* after treatment with the xenobiotic DEHP. *Mj-HSP60* and *Mj-HSP67B2* were highly expressed in the gonad and hepatopancreas, respectively. In addition, these molecules are moderately expressed in the gills, muscle, heart, and stomach. Our findings are consistent with the results of an earlier study showing that the hepatopancreas is the main source of immune

molecules in crustaceans [45]. The hepatopancreas acts as an essential metabolic center in crustaceans and performs versatile roles in defense systems, detoxification, reactive oxygen species production, digestion, absorption, and nutrient secretion. Owing to the critical importance of the hepatopancreas in detoxification and immunological activities, it is highly sensitive to xenobiotic exposure. Similarly, increased upregulation of *HSP90* was noted in the hepatopancreas of *P. monodon* [46]. In addition, three HSPs, namely *MrHSP60*, *MrHSP70* and *MrHSP90*, are constitutively expressed in *M. rosenbergii* during pathogenic infections involving different tissues [47]. Related results were obtained in the Pacific oyster *Crassostrea gigas*, which exhibits highly upregulated *HSP70* expression in the gill tissue after exposure to $Cu^{2+}$ [48]. DEHP has been shown to alter the expression of HSPs in *Chironomus riparius* [49,50]. In this species, *HSP40* and *HSP90* mRNA expression levels increased under various DEHP concentrations for 24 h, which caused morphological deformities [49]. In addition, *HSP70* showed increased expression when treated with low doses of DEHP. Overall, our results indicated that two HSPs, *Mj-HSP60* and *Mj-HSP67B2*, in *M. japonicus* are constitutively expressed, owing to DEHP exposure at day 1. Hence, these molecules can be considered as upregulated responses of xenobiotic levels for the early exposure time in *M. japonicus* crabs. However, at long-term exposure for 7 days, there are different expression patterns between the *Mj-HSP60* and the *Mj-HSP67B2* transcripts. The *Mj-HSP60* expression was downregulated in most crabs after 7 days of DEHP exposure due to reducing cellular immune protection, although expressions of the detoxifying *Mj-HSP67B2* gene [51] were continuously upregulated in DEHP-treated groups compared to the control. *HSP67B2* is significant both in detoxification and in anti-oxidative stress systems, as well as immune protection [26,27,51]. For instance, in *P. trituberculatus*, an important marine and aquaculture species, *Mj-HSP60* displays differential expression patterns in response to environmental salinity stress and exhibits upregulation in the gills [52].

Likewise, *L. vannamei HSP60* mRNA is regulated between 4 and 32 h after the injection of bacteria [53]. *HSP70* is upregulated 24 h after copper exposure in the zebra mussel *Dreissena polymorpha* and midge larvae *Chironomus tentans* [54,55]. In addition, *HSP70* expression is dramatically induced, owing to microbial pathogens in the Chinese shrimp *Fenneropenaeus chinensis* [56]. However, little is known regarding the response of *HSP60* to xenobiotics and stresses in invertebrates such as the sea anemone (*Anemonia viridis*) [29], *D. polymorpha* [54], and the white shrimp (*Litopenaeus vannamei*) [57]. The limited study reported that HSP67B2 acts like a rhodanese homolog with a single RHOD domain, is characterized from the housefly M. domestica, and plays potential roles under oxidative stress conditions [57]. *M. domestica*, and plays potential roles under oxidative stress conditions [51]. In crustaceans, *HSP* expression studies have been conducted on the Asian paddle crab *Charybdis japonica*, with exposure to EDCs (bisphenol A and 4-nonylphenol) [16,58]. To date, this is the first nucleotide and protein sequence information reported regarding *Mj-HSP60* and *Mj-HSP67B2* in the crab species *M. japonicus*. Our gene expression results revealed the potential involvement of the two HSPs in the immune system of crabs. This study highlights the potential importance of these molecules in crustaceans, protecting cells against pathogens as well as in severe cellular and environmental stress conditions.

**Author Contributions:** Conceptualization, K.P., W.-S.K. and I.-S.K; methodology, K.P., W.-S.K. and I.-S.K; formal analysis, K.P., W.-S.K. and I.-S.K; investigation, K.P., W.-S.K. and I.-S.K; resources, K.P., W.-S.K. and I.-S.K; writing—original draft preparation, K.P., W.-S.K. and I.-S.K; supervision, K.P., W.-S.K. and I.-S.K; project administration, I.S.K; funding acquisition, K.P., I.-S.K. All authors have read and agreed to the published version of the manuscript.

**Funding:** This study was supported by the National Research Foundation of Korea, South Korea, which is funded by the Korean Government [NRF-2018-R1A6A1A-03024314] and [NRF-2019-R1I1A1A-01056855].

**Conflicts of Interest:** The authors declare that they have no conflicts of interest.

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
