# Peer review of "Effects of di-(2-ethylhexyl) phthalate on Transcriptional Expression of Cellular Protection-Related HSP60 and HSP67B2 Genes in the Mud Crab Macrophthalmus japonicus"

_applsci, doi:10.3390/app10082766_

Round 1

Reviewer 1 Report

In this manuscript, the authors identified two HSP genes of Japanese mud crab (Macrophthalmus japonicas), HSP60 and HSP67B2, and assessed the effects after exposure to different doses of DEHP. They found the HSP60 and HSP 67B2 were increased significantly response to DEHP toxicity. This study is well-written. However, there are certain issues the authors need to address.

  1. Did DEHP cause any morphological deformities of this crab? The authors should provide phenotypic data.
  2. As the title emphasizes, the highlight of this article is the immune protection through HSP60 and HSP67B2 expression against DEHP toxicity. However, the authors just have qPCR data. The authors need more data to support this strong conclusion, for example, to confirm the protein levels of these two HSPs and/or to assess changes in the immune network around HSP60 and HSP67B2.
  3. How to explain the down-regulated gene expression levels after 7 days of DEHP exposure, especially HSP60? Do the HSPs lose protection from stress after long-term exposure to DEHP?

Author Response

Review1

Comments and Suggestions for Authors

In this manuscript, the authors identified two HSP genes of Japanese mud crab (Macrophthalmus japonicas), HSP60 and HSP67B2, and assessed the effects after exposure to different doses of DEHP. They found the HSP60 and HSP 67B2 were increased significantly response to DEHP toxicity. This study is well-written. However, there are certain issues the authors need to address.

1.Did DEHP cause any morphological deformities of this crab? The authors should provide phenotypic data.

--> To evaluate the potential effects of DEHP exposure on transcriptional expression of Mj-HSP60 and Mj-HSP67B2 genes, we used adult crab individuals involving 3 ± 0.5 cm in shell height, 3.5 ± 0.8 cm in shell width, and 7.5 ± 3.5 g in body weight. Thus, we are not able to observe morphological alteration during crab development induced by DEHP as an endocrine disrupting chemical. For 7 days exposure, there are no significant observation of morphological deformities in the M. japonicus crab exposed to DEHP.

2.As the title emphasizes, the highlight of this article is the immune protection through HSP60 and HSP67B2 expression against DEHP  However, the authors just have qPCR data. The authors need more data to support this strong conclusion, for example, to confirm the protein levels of these two HSPs and/or to assess changes in the immune network around HSP60 and HSP67B2.

--> We totally agree with the above reviewer’s comment. There are no evidence about detailed results about immune network around Mj-HSP60 and Mj-HSP67B2. In the revised manuscript, the title was revised as “Effects of di-(2-ethylhexyl) phthalate on transcriptional expression of cellular protection-related HSP60 and HSP67B2 genes in the mud crab Macrophthalmus japonicus”, because there are only transcriptional expression data of HSP60 and HSP67B2 genes against to DEHP toxicity in the revised manuscript.

3.How to explain the down-regulated gene expression levels after 7 days of DEHP exposure, especially HSP60? Do the HSPs lose protection from stress after long-term exposure to DEHP?

--> At day 1, two HSPs, Mj-HSP60 and Mj-HSP67B2, in M. japonicus are constitutively expressed owing to DEHP exposure. However, at long-term exposure for 7 days, there are different expression pattern between Mj-HSP60 and Mj-HSP67B2 transcripts. The Mj-HSP60 expression was downregulated in the most crabs after 7 days of DEHP exposure as reducing cellular protection, although expressions of detoxifying Mj-HSP67B2 gene were continuously upregulated in DEHP treated groups compared to the control. The downregulation of Mj-HSP60 at day 7 may suggest to possibility of reducing cellular immune protection against to DEHP exposure because HSP60 is a highly immunogenic protein. In addition, HSP67B2 gene is significant both in detoxification and in anti-oxidative stress systems (Tang et al., 2019). Thus the upregulation of Mj-HSP67B2 for 7 days imply detoxifying and antioxidant function as well as immune protection against to DEHP toxicity.

Lines 242-250 on page 6, we have added the above information in the revised manuscript (it indicated as the blue letters shaded by yellow box).

Reviewer 2 Report

The manuscript entitled “Effects of di-(2-ethylhexyl) phthalate on cellular immune protection through HSP60 and HSP67B2 expression in the intertidal mud crab Macrophthalmus japonicus” by Kiyun Park and co-authors reported two newly identified HSP genes, HSP60 and HSP67B2 in the mud crab Macrophthalmus japonicus, and their potential negative immune protection roles in response to DEHP treatment. In general, the experiments are rationally designed, however, the conclusions still need to be carefully revised. I have some concerns below:

Major concerns:

1). The authors tried to conclude DEHP influence the crab immune response, and both HSP60 and HSP67B2 are involved in this immune response pathway. However, there is no evidence provided by the authors showing that DEHP triggers or only triggers an immune response in Macrophthalmus japonicu. Besides, HSP60 and HSP67B2 are involved in many other cellular pathways, except the immune pathway, how the authors conclude the genes expression changes during DEHP treatment are related to immune response.

2). On page 9, line 280, the authors tried to conclude HSP60 and HSP67B2 can be potentially used as biomarkers of xenobiotic levels in Macrophthalmus japonicus crabs. However, the two genes are not constitutively overexpressed after DEHP treatment, both of the genes show a trend to decrease after 7 days. It is hard or impossible to know how many days the crabs in the wild are exposed to DEHP (perhaps always), so it is not applicable to use the relative expression level compared to blank control as a biomarker. The authors should re-consider to choose an internal tissue as a control to potentially use those two genes as DEHP biomarkers. For example, a tissue in which HSP genes are not affected by DEHP treatment, and HSP genes expression level in Gills or Hepatopancreas can be normalized to this tissue.

Minor concerns:

1). The crab samples are directly obtained from a marine market. The authors have limited control before they got those crabs from the market. My concern is the samples will have an artificial bias, for example, those crabs exposed to different food amounts, different stress conditions, different ages, sexes, etc. Did the author consider those bias in their experiments?

2). On page 7, line 216, “HSP60 was expressed approximately 8.2-fold after exposure ...”, should correct to “HSP60 was expressed approximately 8.2-fold higher after exposure…”.

3). How do the DEHP experiment concentrations correlate with the wild environment in the ocean? The authors only give the concentration of DEHP in the sediment, but, how about the surface or bottom water?

4). The authors should label HSP60 and HSP67B2 from M. Japonicus, as MjHSP60, MjHSP67B2, respectively.

5). On page 9, line 286, “In addition, it HSP70 expression is …” may change to “In addition, HSP70 expression is …”.

Author Response

Reviewer 2

Comments and Suggestions for Authors

The manuscript entitled “Effects of di-(2-ethylhexyl) phthalate on cellular immune protection through HSP60 and HSP67B2 expression in the intertidal mud crab Macrophthalmus japonicus” by Kiyun Park and co-authors reported two newly identified HSP genes, HSP60 and HSP67B2 in the mud crab Macrophthalmus japonicus, and their potential negative immune protection roles in response to DEHP treatment. In general, the experiments are rationally designed, however, the conclusions still need to be carefully revised. I have some concerns below:

Major concerns:

1). The authors tried to conclude DEHP influence the crab immune response, and both HSP60 and HSP67B2 are involved in this immune response pathway. However, there is no evidence provided by the authors showing that DEHP triggers or only triggers an immune response in Macrophthalmus japonicu. Besides, HSP60 and HSP67B2 are involved in many other cellular pathways, except the immune pathway, how the authors conclude the genes expression changes during DEHP treatment are related to immune response.

--> HSP60 is well known as a pro-apoptotic molecule, which induces apoptosis and a highly immunogenic protein. HSP67B2 was characterized as a Relish regulated gene in Chinese shrimp innate immunity. HSP67B2 gene is significant both in detoxification and in anti-oxidative stress systems as well as immune function (Tang et al., 2019). However, there is no evidence about immune network around Mj-HSP60 and Mj-HSP67B2. In the revised manuscript, the title was revised as “Effects of di-(2-ethylhexyl) phthalate on transcriptional expression of cellular protection-related HSP60 and HSP67B2 genes in the mud crab Macrophthalmus japonicus”, because there are only transcriptional expression data of HSP60 and HSP67B2 genes against to DEHP toxicity in the revised manuscript.

2). On page 9, line 280, the authors tried to conclude HSP60 and HSP67B2 can be potentially used as biomarkers of xenobiotic levels in Macrophthalmus japonicus crabs. However, the two genes are not constitutively overexpressed after DEHP treatment, both of the genes show a trend to decrease after 7 days. It is hard or impossible to know how many days the crabs in the wild are exposed to DEHP (perhaps always), so it is not applicable to use the relative expression level compared to blank control as a biomarker. The authors should re-consider to choose an internal tissue as a control to potentially use those two genes as DEHP biomarkers. For example, a tissue in which HSP genes are not affected by DEHP treatment, and HSP genes expression level in Gills or Hepatopancreas can be normalized to this tissue.

--> We deleted the word “potential biomarker” as the reviewer’s comment in the revised manuscript. We also revised the content in lines 242-245 and lines 245-250 of the revised manuscript.

Lines 242-245 on page 6; Overall, our results indicated that two HSPs, Mj-HSP60 and Mj-HSP67B2, in M. japonicus are constitutively expressed owing to DEHP exposure at day 1. Hence, these molecules can be considered as upregulated responses of xenobiotic levels for the early exposure time in M. japonicus crabs.

Lines 245-250 on page 6; However, at long-term exposure for 7days, there are different expression pattern between Mj-HSP60 and Mj-HSP67B2 transcripts. The Mj-HSP60 expression was downregulated in the most crabs after 7 days of DEHP exposure as reducing cellular immune protection, although expressions of detoxifying Mj-HSP67B2 gene [47] were continuously upregulated in DEHP treated groups compared to the control. HSP67B2 is significant both in detoxification and in anti-oxidative stress systems as well as immune protection [26, 27, 47].

Minor concerns:

1). The crab samples are directly obtained from a marine market. The authors have limited control before they got those crabs from the market. My concern is the samples will have an artificial bias, for example, those crabs exposed to different food amounts, different stress conditions, different ages, sexes, etc. Did the author consider those bias in their experiments?

--> Actually, all crabs are samples collected by a fisherman in the same area of Yeosu, although we bought crab samples from the market. All crab samples involved 3 ± 0.5 cm in shell height, 3.5 ± 0.8 cm in shell width, and 7.5 ± 3.5 g in body weight. We have to consider an artificial bias in exposure experiment using field samples. To rule out an artificial bias, we divided the crab samples into each experiment group in consideration of size, weight, sex, and health condition.

2). On page 7, line 216, “HSP60 was expressed approximately 8.2-fold after exposure ...”, should correct to “HSP60 was expressed approximately 8.2-fold higher after exposure…”.

--> Page 5, we corrected the lines 191-193 of the revised manuscript as the reviewer’s comment.

3). How do the DEHP experiment concentrations correlate with the wild environment in the ocean? The authors only give the concentration of DEHP in the sediment, but, how about the surface or bottom water?

--> Page 2, we added the concentration of DEHP in the surface or bottom seawater in the lines 43-47 of the revised manuscript.

Lines 43-47 on page 2; In addition, the range of DEHP concentrations was 42-802 ng L-1 and 130-924 ng L-1 in the surface seawater (depth 0.5 m) and bottom seawater (depth 30 m), respectively, in the northwestern Mediterranean Sea [7]. DEHP concentrations were found to range from 62 to 4352 ng L-1 in from bottom to surface seawater of the Bohai Sea and the Yellow Sea, China [8].

4). The authors should label HSP60 and HSP67B2 from M. Japonicus, as MjHSP60, MjHSP67B2, respectively.

--> We revised as Mj-HSP60 and Mj-HSP67B2 in the whole manuscript as the reviewer’s comment.

5). On page 9, line 286, “In addition, it HSP70 expression is …” may change to “In addition, HSP70 expression is …”.

--> Page 6, we revised the line 255 of the revised manuscript as the reviewer’s comment.

Round 2

Reviewer 1 Report

Most of my previous comments have been responded properly so that the manuscript can be accepted.

Reviewer 2 Report

The authors did a great job to resolve all of my concerns. I agree to publish the manuscript as it is presented in the current version.